# A Pilot Study of Multi-Input Recurrent Neural Networks for Drug-Kinase Binding Prediction

**DOI:** 10.3390/molecules25153372

**Published:** 2020-07-24

**Authors:** Kristy Carpenter, Alexander Pilozzi, Xudong Huang

**Affiliations:** Neurochemistry Laboratory, Department of Psychiatry, Massachusetts General Hospital and Harvard Medical School, Charlestown, MA 02129, USA; kcarp@mit.edu (K.C.); apilozzi@mgh.harvard.edu (A.P.)

**Keywords:** artificial intelligence (AI), machine learning (ML), deep learning (DL), recurrent neural network (RNN), virtual drug screening

## Abstract

The use of virtual drug screening can be beneficial to research teams, enabling them to narrow down potentially useful compounds for further study. A variety of virtual screening methods have been developed, typically with machine learning classifiers at the center of their design. In the present study, we created a virtual screener for protein kinase inhibitors. Experimental compound–target interaction data were obtained from the IDG-DREAM Drug-Kinase Binding Prediction Challenge. These data were converted and fed as inputs into two multi-input recurrent neural networks (RNNs). The first network utilized data encoded in one-hot representation, while the other incorporated embedding layers. The models were developed in Python, and were designed to output the IC_50_ of the target compounds. The performance of the models was assessed primarily through analysis of the Q^2^ values produced from runs of differing sample and epoch size; recorded loss values were also reported and graphed. The performance of the models was limited, though multiple changes are proposed for potential improvement of a multi-input recurrent neural network-based screening tool.

## 1. Introduction

Drug development is a long and costly process. A 2014 study found that the cost of developing a prescription drug is, on average, US$2.87 billion [1], and has likely since increased. The beginning of drug discovery, in which a suitable compound to be developed into a drug lead must be found, is particularly slow. The classical approach to this step is called high-throughput screening, and involves repeated in vitro assays of candidate compounds until one with desirable properties is found. High-throughput screening requires all compounds to be purchased or synthesized, and requires repeated menial assaying work on the part of chemists. It also typically has a low yield relative to the amount of required effort. As machine learning grows in popularity, the idea of virtual drug screening has been brought to drug developers. The idea behind virtual screening is to test huge compound databases (either publicly sourced or from private pharmaceutical companies) quickly in silico and pick only the top-scoring compounds to assay in the wet lab. Virtual screening saves researchers time and money, and because it allows for the testing of a much greater number of compounds, it hugely increases yield. There are different forms of virtual screening, with the two primary types being structure-based and ligand-based [2]. The processes for conducting virtual screening also vary, and methods include molecular dynamics simulations (which require enormous amounts of computation) and machine learning.

There exist numerous examples in the literature of the application of various types of machine learning classifiers for virtual screening [3,4,5,6,7,8,9,10,11,12]. Commonly used classifiers include naïve Bayes, k-nearest neighbors, support vector machines, random forests, and artificial neural networks (ANNs). ANNs in particular are often superior to other methods in various applications, and outperform other classifiers in certain virtual screening applications [13]. In this report, we describe and evaluate a machine learning approach to the prediction of binding activity between potential drugs and their protein kinase targets. Our approach is centered around a multi-input recurrent neural network (RNN), using one-hot and embedded feature representation. RNNs are notable in that they allow cycles between layers, and they incorporate information from previous inputs in predicting the current input. RNNs are commonly used in linguistic applications, as they are ideal when inputs are sequential and dependently related [14]. Within the fields of biology and chemistry, RNNs have been employed for generating molecular structures and features [15,16].

Protein kinases are desirable drug targets as they are central to the activation or deactivation of numerous other proteins in the cell through phosphorylation [5]. They comprise one of the largest drug target families [17], and are a central focus in the treatment of various cancers, inflammatory diseases, and central nervous system disorders, among other ailments [18]. Many diseases are caused not by the presence of a novel protein, but rather by irregular expression of an existing protein. It therefore makes sense to inhibit abnormal levels of protein activation, rather than general protein functioning [19]. Additionally, the phosphorylation of proteins is typically a point of amplification in a biological signaling pathway—that is, one activated protein kinase can phosphorylate many protein targets. Therefore, it is more effective to inhibit the disease pathway further upstream than at the level of the protein directly causing the disease phenotype, simply because there are fewer protein kinases than there are phenotype-affecting proteins downstream.

There are multiple ways of measuring binding activity. In this study, we used the *IC*50 metric, which reports the concentration of a compound required to reach half the maximal inhibitory activity. In order to develop our models, data made available through the IDG-DREAM Drug-Kinase Binding Prediction Challenge were used. The challenge, which focused on the crowdsourced development of kinase-inhibitor prediction models, provides access to a variety of molecule-interaction databases [20].

## 2. Results

### 2.1. Raw IC50 Values as Output

The first models implemented were trained to predict the raw IC_50_ values. In the training set of 3000 samples, the IC50s ranged from 1 × 10^−3^ to 2.75 × 10^11^ nM, with an average value of 9.45 × 10^7^ nM. Out of nine cross-validation attempts, four finished with nonzero, non-NaN (not-a-number) values. A mix of simplified molecular-input line-entry system (SMILES) and IUPAC International Chemical Identifier (InChI) representations were tried; all SMILES implementations resulted in predictions consisting entirely of zeros or NaNs. The Q^2^ values were all hugely negative (Q^2^
*<* 10,000), indicating performance much worse than random.

The distribution of one such nonzero cross-validation IC50 prediction was analyzed: it had a minimum of 1.84 × 10^11^ nM, a maximum of 1.52 × 10^12^ nM, and an average of 8.02 × 10^11^ nM.

### 2.2. log IC50 Values as Output

A second iteration of model tuning used data identical to those of the first iteration, except with the logarithms of the IC50s as output. Four models were evaluated with this metric (Table 1). It is worth noting that the models with similar numbers of training examples and similar amount of training time behaved similarly, whereas models with different input representations did not necessarily perform differently.

For further validation, the Q^2^
_F3_ metric, which can be a more reliable measure than Q^2^ alone, was computed as described by Todeschini et al. [21,22]. The value was computed based on five different cross validation runs, using 10,000 samples split into five folds (~8000 training and ~2000 testing per run) and 200 epochs for the one-hot and embedding models. The Q^2^ and Q^2^
_F3_ values for each model are shown in Table 2.

The loss was plotted for some of these runs, and can be found in Figure 1, Figure 2 and Figure 3.

### 2.3. Baseline Performance

The baseline model was also trained on 3000 samples for 500 epochs. In 5-fold cross-validation, it obtained the following Q^2^ scores: [−0.3117, −0.3956, −0.4183, −0.5979, −0.3563]. Its loss plots generally showed convergence after 100 epochs (Figure 3). The mean Q^2^ value of five additional cross-validation runs with the same parameters was −0.4959. The mean Q^2^_F3_ value of the five additional runs was found to be −0.4928.

## 3. Discussion

Our primary interpretation of results is grounded in the meaning of the R2 metric. A perfect model will have R^2^ = 1. A model that deterministically outputs the mean of the expected distribution will have R^2^ = 0. The more negative an R^2^ value, the worse it is. In the case of quantitative structure-activity related models, R^2^ typically refers to the predictions made by the model on items from its training dataset, while predictions made on external data not used for training relate to the Q2 value. In this work, we report the Q^2^ as the primary measure of model fit. Q^2^_F3_ values were also computed for each of the model types and were relatively close to the reported Q^2^ for each model.

With this in mind, we can judge all models using the raw IC50 values as completely ineffective, as their Q^2^ values were much less than zero. Because the models using SMILES strings as compound input representation could not generate real and nonzero predictions, we continued experiments under the assumption that using the InChIs would have better performance.

When the log IC50 was used instead, we see a dramatic improvement in Q^2^. The two models that surpassed the Q^2^
*>* 0 threshold both had a training set with 10,000 samples, implying that having more training data improves performance. For each of the two model pairs that have the same overall setup but different input representations (i.e., one uses one-hot encoding and the other uses embedding), neither model in the pair has obviously better Q^2^ values than the other. This implies that the input representation does not have a large effect on performance.

The results from the baseline model are similar to those of the recurrent models that were also trained for 500 epochs on training sets with 3000 samples. We interpret this as meaning that, at the 3000-sample level, the use of recurrence does not aid the model.

Examination of loss plots yields more insight. The loss plots for RNNs trained on 3000 samples were generally very noisy and divergent (Figure 1).

This is indicative of their subpar performance in terms of Q^2^. Meanwhile, the loss plots for RNNs trained on 10,000 samples show progress toward convergence (Figure 2).

The fact that convergence has not been definitively reached implies that training for more epochs may further improve performance. The baseline loss plot (Figure 3) appears to converge around 100 epochs, implying that training the model longer will not affect Q^2^. The fact that the RNN model has room for improvement and the baseline does not is perhaps indicative of value being gained from the complexity of recurrence. The data used to train the models were structural in nature, meaning with the order of the features was relative to their respective position in the sequence of molecules or amino acids. A simple neural network, as was used for the baseline, would not properly account for the ordering of each sample’s features, greatly hampering its accuracy.

### Future Work

It has not escaped our attention that none of our proposed models have particularly desirable Q^2^ values. A regression model that deterministically outputs the mean of the expected distribution has Q^2^ = 0, and because our attempts have negative or small positive Q^2^ values, we must concede that our approach has flaws. However, this does not mean that it is impossible for a multi-input RNN to achieve good performance on the drug-kinase binding problem. We suggest the following experiments for model improvement in a setting where time is not as much of a constraint:

The hyperparameters of the proposed models mostly stayed fixed throughout the tuning process; it is important to vary them to ensure model optimization. Future work would involve conducting grid search or random sampling over a number of hidden nodes in recurrent and dense layers, batch sizes, and learning rates. Performance may also improve with an increased training set size. There are hundreds of thousands of viable datapoints that were left unused—training the models on more data does not require more procurement work on the part of the researcher, only more computing time. The method of input representation can also be refined. Further experiments should continue to explore both the one-hot model and the embedding model in an effort to discover if one is preferable. More work should also be done in exploring SMILES-based models and either confirming or disproving that InChIs contain more useful latent information. With regard to proteins, more work needs to be done in comparing sets of features used to describe the amino acids and determining which biochemical properties are most useful to include for this particular problem. One other potential extension of this work would be to attempt using a more complex model. For example, a graph convolutional neural network (GCNN) may fare well in approaching the drug–target binding problem due to its ability to capture the physical nature of molecules.

## 4. Methods

### 4.1. Data

#### 4.1.1. Data Collection

A dataset of 3,717,236 experimentally derived compound–target interactions was obtained from the IDG-DREAM Drug-Kinase Binding Prediction Challenge. The information given with each sample included the ChemBL identifier of the compound, the UniProt identifier of the target, and a quantification of their binding activity. Binding was described as a K_D_, K_I_, or IC_50_ value. For consistency, only datapoints with binding activity given as an IC_50_ value in nM were considered.

As compounds and their targets are ultimately molecules governed by the laws of physics and chemistry, a deep learning model will likely benefit from the incorporation of biochemical information. Fortunately, there already exist standardized string representations of molecular compounds that have chemical significance. The most commonly accepted and used are the simplified molecular-input line-entry system (SMILES) string [23] and the IUPAC International Chemical Identifier (InChI) [24]. Both systems encode the constituent atoms of a molecule, in addition to how those atoms are connected to each other.

While small molecule drug compounds have SMILES strings and InChIs of feasible length (for this dataset: average SMILES length = 55 characters; average InChI length = 155 characters), protein kinases are a different class of biological molecule. Protein kinases are proteins built from a sequence of amino acids, each of which consists of dozens of atoms. There are

There are 20 naturally occurring amino acids that cells use to synthesize proteins. Each amino acid has a standardized single-letter code, meaning that proteins can be represented as a string where each character at a given location represents an amino acid. This is a shorter and more informative representation than a large SMILES or InChI.

The chemical representations described above were generated with the aid of the Requests library [25] for making API calls. The SMILES and InChI representations of each compound were queried from the ChemBL database (European Molecular Biology Laboratory, Cambridge, UK) [26], and the primary sequence of each target was queried from the UniProt database (European Molecular Biology Laboratory, Cambridge, UK and the Swiss Institute of Bioinformatics, Lausanne, Switzerland) [27].

#### 4.1.2. Data Representation

When SMILES strings or InChIs were fed into a network as inputs, they were first converted to a one-hot representation. This is to remove any false correlation between characters in the strings that are close in American Standard Code for Information Interchange (ASCII) encoding. This transformed a string of characters into a list of vectors, with each vector consisting of all zeros and a single one at the index of the character it represents. There is precedent in the literature to use a one-hot encoding for SMILES strings and InChIs [6]. Input vector lengths were standardized to be 100 characters for SMILES and 200 characters for InChI; these numbers were chosen based on the study conducted in [6]. Shorter sequences were padded at the beginning with zero-vectors, and longer sequences were truncated.

It is possible to use a similar embedding method for the primary sequences of proteins, and many machine learning classifiers in the literature do so [7,8]. However, this disregards the fact that amino acids can be partitioned into groups with similar charge or steric bulk, and that the single-letter encoding (derived from the name of each amino acid) does not allow for easy distinction between these groups. For example, isoleucine (I) and leucine (L) both have small, hydrophobic side chains, and even have identical chemical formulas. However, if the list of single-letter amino acid encodings is given alphabetically (as it often is), the amino acid lysine (K) falls between isoleucine and leucine. As lysine is extremely different from both isoleucine and leucine (it has a long, positively-charged side chain), any system of protein representation that creates an implicit distance between amino acids based on their single-letter encoding will introduce inherent biochemical misinformation. At best, a one-hot encoding of the amino acid set can be used—but while it does not encode misinformation, it also loses biochemical knowledge that is freely available.

For this reason, we represent each amino acid in a given protein kinase primary sequence by a vector of biochemical properties. Seventeen features (Table 3) were chosen on the basis of general biological importance as well as having been found in previous studies [9,10,11] to be useful in characterizing protein activity. The values for each of these features, for each amino acid, were taken from the AAindex database (Kyoto University Bioinformatics Center, Kyoto, Japan) [28], then scaled to range between −1 and 1.

Similar to the compound representation, the length of the amino acid sequence vectors was fixed to be 500 feature vectors long (meaning that the shape of the overall representation for a given sequence was (500, 17)). Shorter proteins were padded at the beginning with zero-vectors of length 17, and longer proteins were truncated.

### 4.2. Network Architectures

#### 4.2.1. Multi-Input Recurrent Neural Network with One-Hot Encoding

The principle architecture proposed by this project is an augmentation of the standard, simple RNN (Figure 4).

The problem of compound–target binding inherently has two input variables: the compound and the target. It is important to take both of these into account, since intuitively one compound will not have the same binding activity with all targets, and vice versa. We cannot simply concatenate the representations of compound and target, because the RNN structure would consider them as one sequence. This would be a misrepresentation of the spatial nature of binding, in addition to having the potential for placing more importance on whichever input comes second (since an RNN may ”forget” elements that come early in a long sequence). Therefore, it is better to first consider both inputs separately, then combine their learned representations to draw conclusions about their binding. We accomplish this through the RNN architecture shown below. The one-hot representation of the compound, generated as described previously, is fed into a recurrent layer. Likewise, the one-hot representation of the target is fed into another recurrent layer. These two layers are trained separately. The compound recurrent layer had about half as many hidden nodes as there were possible characters for its given string representation. For example, there were 46 distinct characters present in all InChIs used, so the compound recurrent layer for InChI models used 22 hidden nodes. The target recurrent layer also followed this rule of thumb. Since there are 20 possible amino acids, this layer was initialized with 12 hidden nodes. The outputs of the two recurrent layers are combined through concatenation. A fully connected hidden layer with 7 hidden nodes and rectified linear unit (ReLU) activation performs a nonlinear transformation, then feeds the data representation into the output layer, which consists of a single unit and ReLU activation.

#### 4.2.2. Multi-Input Recurrent Neural Network with Embedding

In addition to the primary proposed model, we also constructed models that have a different method of input representation. As an alternative to the one-hot representations, embedding layers were used (Figure 5).

To prepare the string input for the embedding layer, each character in the string was converted to an index. For example, each of the 20 amino acids were mapped to an index such that a protein sequence in the form of a string of letters was converted to a list of integers ranging from 0 to 19. The embedding layer takes that vector and learns weights to transform it to an abstracted representation with a new dimension. In this case, the embedding layers were set to output a length-64 vector representation. These outputs were concatenated, then fed into a hidden layer with 32 nodes, and finally a single output node. As before, all layers used ReLU activation functions.

#### 4.2.3. Baseline

A simple neural network architecture (Figure 6) was created, trained, and evaluated for purposes of comparison. This network maintains the separation of compound and target inputs, but replaces the recurrent layers with dense layers and uses embedding instead of one-hot encoding. The embedding layers transform the inputs into vectors with a length of 64, which are then concatenated together and fed into a hidden layer with 32 nodes. The resulting arrays were flattened before being fed into the final output node. All dense layers used ReLU activation functions.

### 4.3. Implementation

All models were written in Python 2.7.11 (Python Software Foundation, Wilmington, DE, USA) using Keras [29] libraries and run on the C3DDB (Commonwealth Computational Cloud for Data Driven Biology) computing cluster. Final models were trained for 500 epochs with a batch size of 32. An Adam optimizer was used with a learning rate of 0.001. Mean squared error was used as the loss function.

In the initial training phase, models were given 3000 datapoints. After processing these data as described in Section 2.1, they were partitioned into 5 random splits for 5-fold cross-validation. In each iteration of cross-validation, the model was trained on four partitions, then made predictions on the fifth. The predicted values were compared to the real values using an *R*^2^ test from the scikit-learn library [30].

Final models were produced by training using 10,000 datapoints. Again, 5-fold cross-validation was used, for the purpose of final tuning and selection of the optimal model. In effect, each run of validation involved ~8000 training and ~2000 testing datapoints.

## 5. Conclusions

Overall, it would appear that the simple RNN-based approach for predicting the inhibitive relationship between compounds and a protein target was ineffective. Though further investigation is warranted, it may also be that RNNs are ill-suited to this value-prediction task, particularly given a dataset that is disparate, with samples that may not be related enough for the advantages of a recurrent network to be fully realized; as mentioned previously, a convolutional neural network (CNN) model such as a graph CNN (GCNN) model may prove to be more effective. However, the poor performance of both the RNN and baseline models indicates that there are likely other issues with the present approach. It is likely that the limited compound and target structural data used were insufficient for accurately assessing the relationship between them. More descriptive structural data beyond that of simple SMILES or INCHI strings and amino-acid sequences/features would likely improve performance; the inclusion of important components of molecular interaction, such as binding site information, is perhaps key to further development a successful model. Several avenues exist for the development of a more successful approach based on the findings of our pilot study, which can be explored in future work.

## Figures and Tables

**Figure 1 molecules-25-03372-f001:**
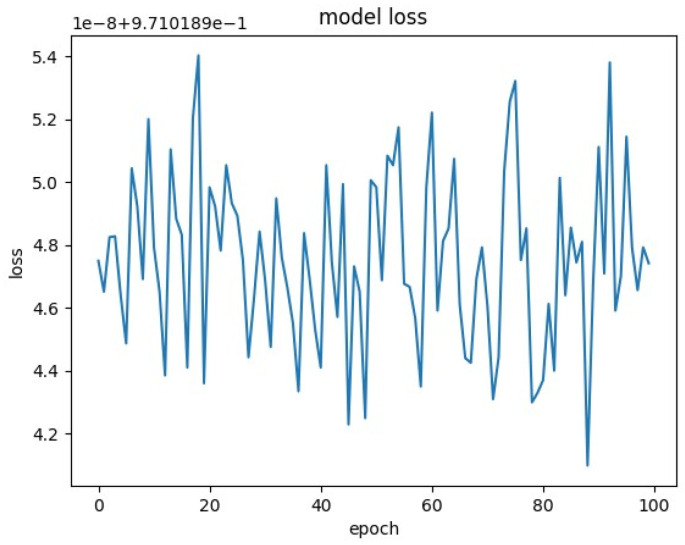
Loss plot for model with 3000 samples, 500 epochs, one-hot encoding of IUPAC international chemical identifiers (InChIs).

**Figure 2 molecules-25-03372-f002:**
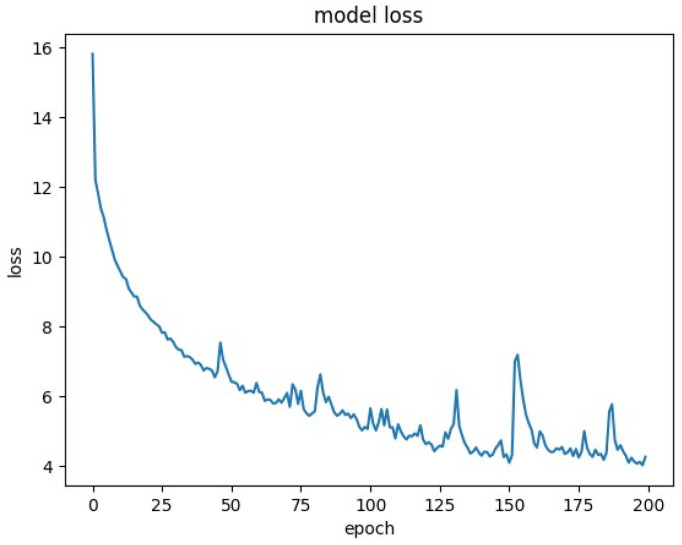
Loss plot for model with 10,000 samples, 200 epochs, embedding of InChIs.

**Figure 3 molecules-25-03372-f003:**
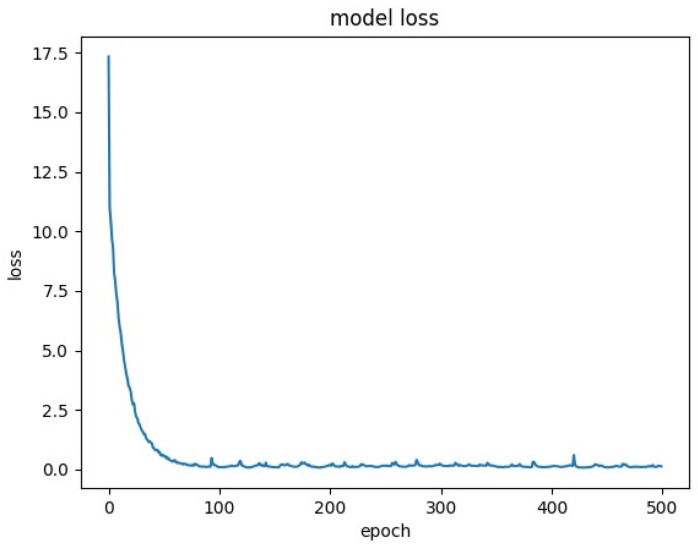
Loss plot for baseline model with 3000 samples, 500 epochs.

**Figure 4 molecules-25-03372-f004:**
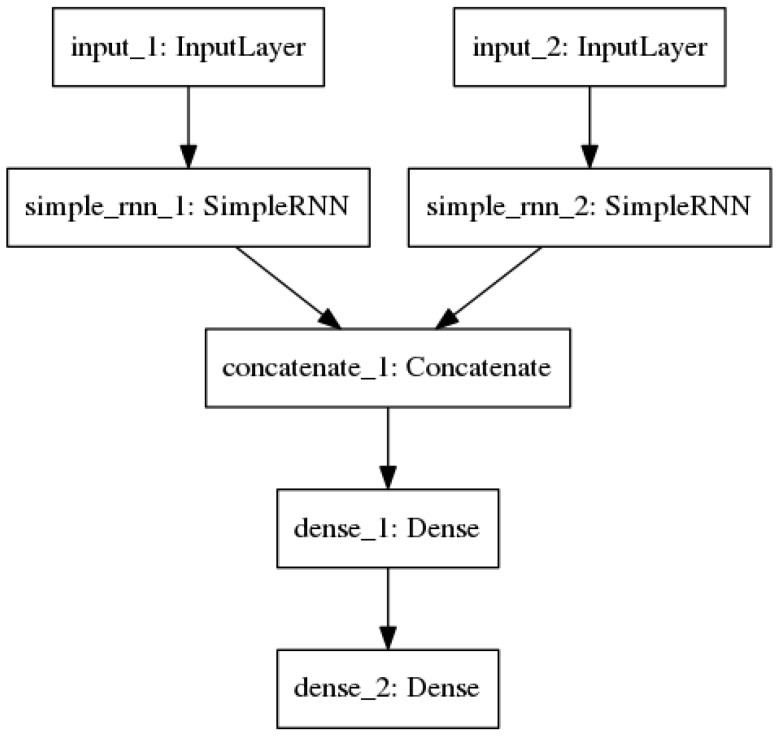
The multi-input recurrent neural network (RNN) architecture with one-hot encoded inputs.

**Figure 5 molecules-25-03372-f005:**
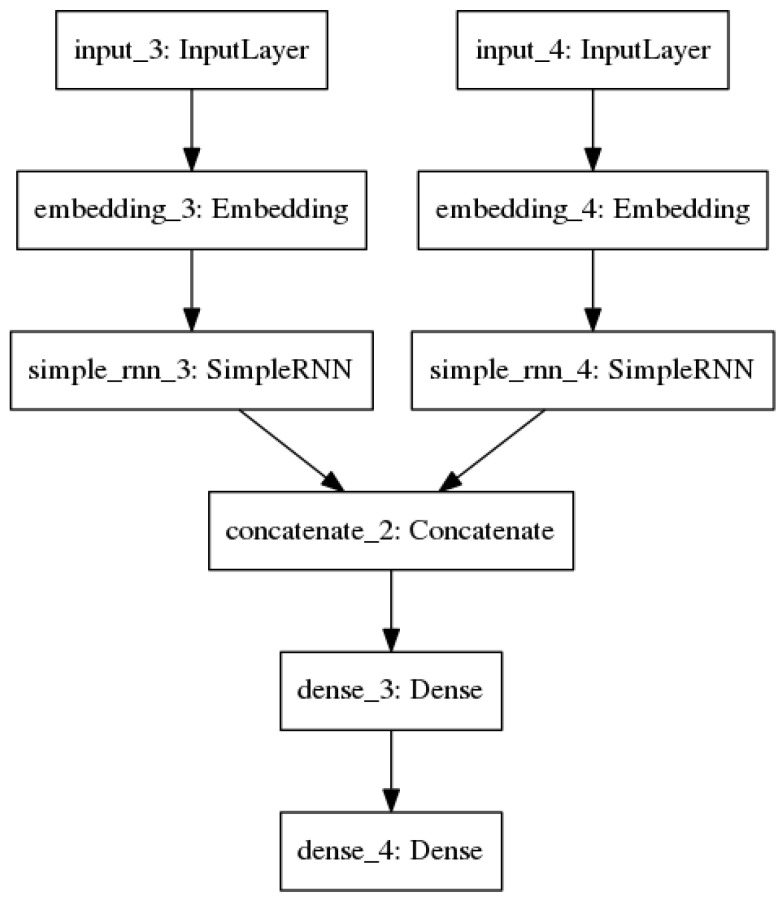
The multi-input RNN architecture with string inputs that undergo embedding.

**Figure 6 molecules-25-03372-f006:**
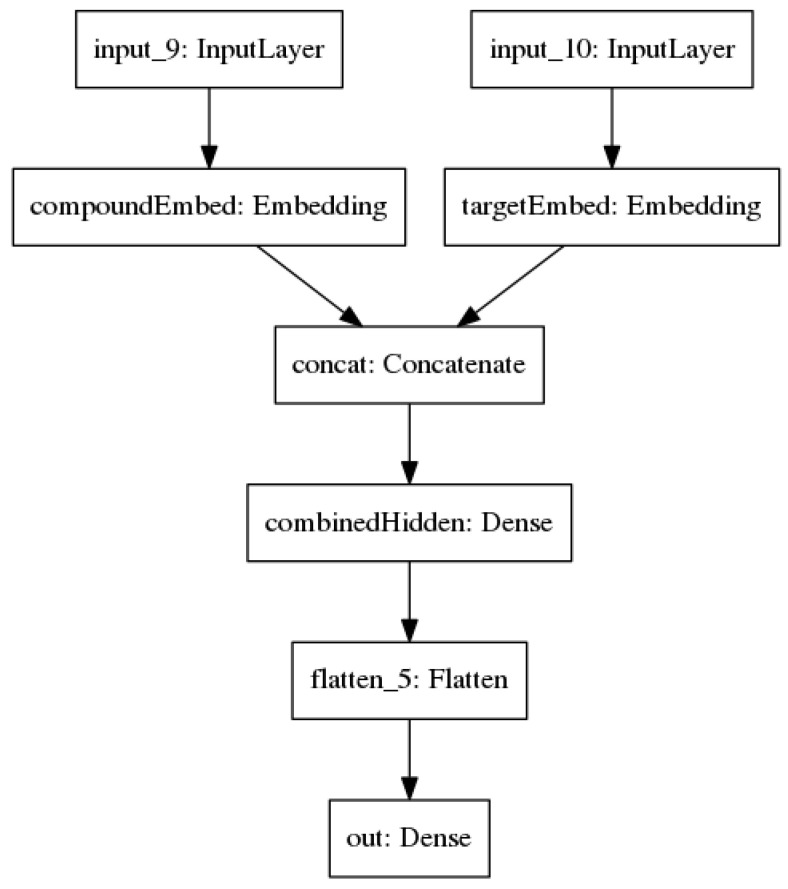
The baseline neural network (NN) architecture with string inputs that undergo embedding.

**Table 1 molecules-25-03372-t001:** Q^2^ values for various models with log IC_50_ output.

Samples	Epochs	Input Rep.	[Q^2^ for x-val]
3000	500	One-hot	[−3.8677]
3000	500	Embed	[−0.3743, −0.3382, −0.3323, −0.3917, −0.2402]
10000	200	One-hot	[0.0510, 0.0005, 0, 0.0671]
10000	200	Embed	[0.0391, 0, 0.0364, −0.0689, −0.0512]

**Table 2 molecules-25-03372-t002:** Q^2^ values and Q^2^
_F3_ values for various models with log IC_50_ output.

Model/Input Rep.	Mean Q^2^	Mean Q^2^_F3_
One-hot	0.0601	0.0714
Embed	0.0537	0.0467

**Table 3 molecules-25-03372-t003:** Biochemical features for amino acids in protein targets.

AAindex ID	Description
KLEP840101	Net charge
KYTJ820101	Kyte-Doolittle hydrophobicity
FASG760101	Molecular weight
FAUJ880103	Normalized Van der Waals volume
GRAR740102	Polarity
CHAM820101	Polarizability
JANJ780101	Average accessible surface area
PRAM900102	Relative frequency in alpha-helix
PRAM900103	Relative frequency in beta-sheet
PRAM900104	Relative frequency in reverse-turn
NADH010104	Hydropathy scale (20% accessibility)
NADH010107	Hydropathy scale (50% accessibility)
RADA880103	Transfer free energy from vap to chx
RICJ880113	Relative preference value at C2
RICJ880112	Relative preference value at C3
KHAG800101	Kerr-constant increments
PRAM820103	Correlation coefficient in regression analysis

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
