# Peer review of "A Pilot Study of Multi-Input Recurrent Neural Networks for Drug-Kinase Binding Prediction"

_molecules, 2020, doi:10.3390/molecules25153372_

Round 1

Reviewer 1 Report

Application of artificial intelligence algorithm in drug design area is a very hot area right now. And the topic of this MS is very interesting and the concept is eminently achievable, especially, the authors chose the kinase system as the targets, which has rich data for training the model. But I still have some questions needed to be addressed.

  1. Kinase is a very well-studied drug target and therefore, there should be a very large experimental compound-target data set available for training the model. According to the MS, I noticed that the MS only focused on IDG-DREAM, I suggest to try to collect more data to make the method more powerful. Please refer to the experimental database, such as BindingDB, etc.
  2. There are several AI algorithms, why you chose RNNs?
  3. How did you consider about the binding mode of the ligand-protein? It is very important for determining the selectivity for drugs to different but structurally and sequentially similar kinases. Therefore, the selectivity for Kinase is the most challenging problem. In other words, discrimination for the selectivity is the one criteria for evaluating a kinase relating method.
  4. How did you evaluate quality of your model? i.e., how do you know your model is not over fitted?
  5. Will you provide your codes to the public?

Author Response

Reviewer 1’s critiques and Responses

Application of artificial intelligence algorithm in drug design area is a very hot area right now. And the topic of this MS is very interesting and the concept is eminently achievable, especially, the authors chose the kinase system as the targets, which has rich data for training the model. But I still have some questions needed to be addressed.

  1. Kinase is a very well-studied drug target and therefore, there should be a very large experimental compound-target data set available for training the model. According to the MS, I noticed that the MS only focused on IDG-DREAM, I suggest to try to collect more data to make the method more powerful. Please refer to the experimental database, such as BindingDB, etc.

Response: This work was intended as a pilot study for development of RNN-based screening tools. We plan to include more diverse datasets in future work.

  1. There are several AI algorithms, why you chose RNNs?

Response: Though RNNs have seen use in compound-generation problems, there has been little work done on their applications in problems of binding affinity value prediction in the context of ligand-target binding.

  1. How did you consider about the binding mode of the ligand-protein? It is very important for determining the selectivity for drugs to different but structurally and sequentially similar kinases. Therefore, the selectivity for Kinase is the most challenging problem. In other words, discrimination for the selectivity is the one criteria for evaluating a kinase relating method.

Response: The models we developed were trained on fairly simple compound-structure and target-structure datasets that did not account for binding modes. The overall performance of our models indicates that more detailed data, such as binding-modes, pocket-structure, etc. will be helpful. We plan to integrate such information in our future models.

  1. How did you evaluate quality of your model? i.e., how do you know your model is not over fitted?

Response: Our assessments of the accuracies of our models were primarily based on the relationship between our predictions and the true values, as measured through the R2 values for each set of predictions made.

  1. Will you provide your codes to the public?

Response: We will make the code available upon request.

Reviewer 2 Report

The manuscript is quite interesting but it presents some points to be addressed:

Authors should better explain the kind of tests performed in the data collected by the database and the targets enclosed in this database

the proposed approach although interesting should be further validated in order to assess the performance of the approach. there is no mention about a test set for validating the proposed approach. In my opinion in the manuscript a section with the validation methods should be added reporting a validation at least in silico of the proposed approach. In other terms, if I have a group of molecules, the model is able to predict if in the group is present a kinase inhibitor? and which kind of kinase could be the target of compound

Author Response

Reviewer 2’s critiques and Responses

The manuscript is quite interesting but it presents some points to be addressed:

Authors should better explain the kind of tests performed in the data collected by the database and the targets enclosed in this database.

Response: The databases mentioned contain information on a variety of assays pooled from different primary literature sources; we specifically pulled information regarding IC50 assays only, as that was the focus of our model. Data collection is described in the methods section.

the proposed approach although interesting should be further validated in order to assess the performance of the approach. there is no mention about a test set for validating the proposed approach. In my opinion in the manuscript a section with the validation methods should be added reporting a validation at least in silico of the proposed approach. In other terms, if I have a group of molecules, the model is able to predict if in the group is present a kinase inhibitor? and which kind of kinase could be the target of compound.

Response: Though we did not have explicit training and tests sets that were predefined, the data was separated into training and testing sets for cross-validation.

Reviewer 3 Report

  1. There is a lack of conclusion section.
  2. Reference format is inappropriate.

Author Response

Reviewer 3’s critiques and Responses

  1. There is a lack of conclusion section.

Response: We have added a conclusion section.

  1. Reference format is inappropriate.

Response: The references were structured based on the reference style specifications instructed by the journal. We have made corrections if needed.

Round 2

Reviewer 2 Report

The revised version of the manuscript only marginally considered my previous comments

"the proposed approach although interesting should be further validated in order to assess the performance of the approach. there is no mention about a test set for validating the proposed approach. In my opinion in the manuscript a section with the validation methods should be added reporting a validation at least in silico of the proposed approach. In other terms, if I have a group of molecules, the model is able to predict if in the group is present a kinase inhibitor? and which kind of kinase could be the target of compound."

for validating a model is not sufficient to assess the performance using cross-validation methods. The model should validated by using:

-external test set not used for building the model

-decoys set for assessing the performance of the model

-Q2F3 metrics in addition of the r2 and /or Q2. For Q2F3 please check the article of Todeschini et al https://pubs.acs.org/doi/abs/10.1021/acs.jcim.6b00277

-experimental validation about potential kinase inhibitors retrieved by using the developed model

without the step of validation the manuscript cannot be considered for publication in high rank journal like this.

Author Response

Reviewer 2’s critiques and Responses (2nd round)

The revised version of the manuscript only marginally considered my previous comments

"the proposed approach although interesting should be further validated in order to assess the performance of the approach. there is no mention about a test set for validating the proposed approach. In my opinion in the manuscript a section with the validation methods should be added reporting a validation at least in silico of the proposed approach. In other terms, if I have a group of molecules, the model is able to predict if in the group is present a kinase inhibitor? and which kind of kinase could be the target of compound."

for validating a model is not sufficient to assess the performance using cross-validation methods. The model should validated by using:

-external test set not used for building the model

Response: For all assessments of performance, the data was made on a subset of our overall dataset that was not used in training that particular model. No predictions made on training data were considered in validation. Looking into this further, it seems we should have used the term Q2 rather than R2. We have corrected this throughout the work.

-decoys set for assessing the performance of the model

Response: In this pilot study, we believe further validation of a decoy dataset would not be worthwhile at this time. In future iterations, we agree that would be a useful assessment alongside experimental validation.

-Q2F3 metrics in addition of the r2 and /or Q2. For Q2F3 please check the article of Todeschini et al https://pubs.acs.org/doi/abs/10.1021/acs.jcim.6b00277

Response: We have calculated & added measures of the Q2F3 metric for each model type.

-experimental validation about potential kinase inhibitors retrieved by using the developed model

Response: Experimental validation would indeed be an essential part of future work; this is intended as pilot study exploring the potential of an RNN-based model.

without the step of validation the manuscript cannot be considered for publication in high rank journal like this.

Response: We have added further validation and made it more clear that the reported measures are not based on predictions made on the training data.